# Underwater Hyperspectral Imaging (UHI): A Review of Systems and Applications for Proximal Seafloor Ecosystem Studies

Juan C. Montes-Herrera [1,*], Emiliano Cimoli [1,2], Vonda Cummings [3], Nicole Hill [1], Arko Lucieer [2] and Vanessa Lucieer [1]

1   Institute for Marine and Antarctic Studies, University of Tasmania, Tasmania 7001, Australia; emiliano.cimoli@utas.edu.au (E.C.); nicole.hill@utas.edu.au (N.H.); vanessa.lucieer@utas.edu.au (V.L.)
2   Discipline of Geography and Spatial Sciences, School of Technology, Environments and Design, College of Sciences and Engineering, University of Tasmania, Tasmania 7001, Australia; Arko.Lucieer@utas.edu.au
3   National Institute of Water and Atmospheric Research, Wellington 14901, New Zealand; vonda.cummings@niwa.co.nz
*   Correspondence: juancarlos.montesherrera@utas.edu.au; Tel.: +61-0458368069

**Abstract:** Marine ecosystem monitoring requires observations of its attributes at different spatial and temporal scales that traditional sampling methods (e.g., RGB imaging, sediment cores) struggle to efficiently provide. Proximal optical sensing methods can fill this observational gap by providing observations of, and tracking changes in, the functional features of marine ecosystems non-invasively. Underwater hyperspectral imaging (UHI) employed in proximity to the seafloor has shown a further potential to monitor pigmentation in benthic and sympagic phototrophic organisms at small spatial scales (mm–cm) and for the identification of minerals and taxa through their finely resolved spectral signatures. Despite the increasing number of studies applying UHI, a review of its applications, capabilities, and challenges for seafloor ecosystem research is overdue. In this review, we first detail how the limited band availability inherent to standard underwater cameras has led to a data analysis "bottleneck" in seafloor ecosystem research, in part due to the widespread implementation of underwater imaging platforms (e.g., remotely operated vehicles, time-lapse stations, towed cameras) that can acquire large image datasets. We discuss how hyperspectral technology brings unique opportunities to address the known limitations of RGB cameras for surveying marine environments. The review concludes by comparing how different studies harness the capacities of hyperspectral imaging, the types of methods required to validate observations, and the current challenges for accurate and replicable UHI research.

**Keywords:** imaging spectroscopy; marine pigments; benthic habitat; remotely operated vehicle (ROV); spectral analysis; mapping; seafloor

## 1. Introduction

### 1.1. Background

The rapid and extensive effects of anthropogenic activities on marine seafloor ecosystems range in scales from global to local to individual organisms [1–3]. The state of marine ecosystems is accelerating towards a similar tipping point in biodiversity and ecosystem functioning first observed in terrestrial ecosystems before the Industrial Revolution [4,5]. The consequences of such a transition in ecosystem services provided to human communities are still uncertain [6–8]. Ecosystem-based management requires cost-efficient monitoring methods that guarantee accurate observations about the state and functioning of seafloor ecosystems and that are capable of synthesizing information at multiple spatial and ecological scales [9–11].

Satellite and air-borne (drone or aircraft) imaging spectroscopy at high spectral resolutions (<10 nm bands), also referred to as hyperspectral imaging, has been advantageous for

collecting observations over broad areas (100 s m$^2$ to 100 s of km$^2$) about the extent and condition of different types of coastal marine ecosystems (<15 m depth) [11–13]. Imaging spectroscopy, in general, aims to obtain the spectrum for each pixel in the image of a scene to classify objects, identify materials, or detect and quantify processes [14,15]. Multi- and hyperspectral applications over coastal marine ecosystems have proven to be an efficient and accurate tool as they have enabled detailed benthic features to be mapped, such as benthic habitat type, macro- and micro-algae cover, coral health, and structural forms [16,17]. Yet, it remains challenging to accurately classify targets that have similar spectral signatures (e.g., bleached coral vs. white sand, macroalgae species), and a limitation is that deeper ecosystems (>10 m) are neglected [18–20].

Most of the ocean is in fact "optically deep" to satellite or aerial platforms (i.e., the signal from the substratum is insignificant or undetectable). This is of particular concern for unexplored seafloor ecosystems where human impacts are pervasive yet can go unnoticed, such as: (a) mesophotic reef systems (found from 40 m to 150 m in depth), which in some areas, extend for approximately 2000 km and are being discovered in every ocean [21]; (b) deep ocean habitats (up to 3000 m depth) that cover a total 50% extent of the world's ocean with less than 0.01% being sampled or studied [1,22]; and (c) polar oceans, where seasonal darkness or ice cover limit measurements and human observations are restricted or impossible [23,24].

Proximal (or close-range) imaging spectroscopy (from 1 to 150 m distance from the target) can provide imagery at high spatial (from 1 to 100 mm) and spectral (from 1 to 15 nm) resolutions [25]. Within the past decade (since 2013), marine ecosystem researchers have tested taking hyperspectral imaging systems underwater using waterproof enclosures referred to as underwater hyperspectral imaging (UHI) [26]. In particular, UHI has shown promise as a bio-optical tool for automated identification of benthic organisms, biogeochemical features, and habitat classification [26,27]. However, along with the range of opportunities that this new methodology provides, there are also technical challenges associated with working in an optically complex and difficult-to-access underwater environment that still requires considerable research effort. Over the past five years, UHI's adaptability to marine surveying has witnessed extensive progress through development in several seafloor ecosystem studies (Figure 1).

Studies of marine ecosystems are increasingly embracing or proposing UHI to be able to deliver ecological information of diverse biogeochemical processes across multiple spatial scales (from mm$^2$ to 100 s km$^2$) at high spectral resolutions, in a non-invasive manner, opening possibilities to relate the measurements to other attributes in the immediate environment [28,29]. Appropriate monitoring of seafloor ecosystems calls for the development of underwater proximal observations, also referred to as "close-range" observations, (~1–5 m distance from the seafloor) capable of covering geographical extents from small to broad areas of the seafloor. Recent proximal applications of hyperspectral imaging of seafloor ecosystems reveal small-scale patterns (~mm$^2$ to m$^2$) that would not have been recognized in previous broad scale aerial observations (Figure 1). Understanding how global environmental changes impose selective pressures on the local and individual scale and modify ecosystem processes, such as productivity, organism interactions and recruitment, and nutrient cycling, is required not only for determining species biodiversity but also for integrating the complexities of ecosystem functioning and environmental change [2,30]. In the next section of this review, we present an in-depth analysis of the role of traditional underwater imaging for seafloor ecosystem studies, demonstrating how the lack of spectral resolution has led to a "bottleneck" in seafloor research. We explore the increasing need for hyperspectral resolution to automate benthic classification and increase our monitoring capabilities.

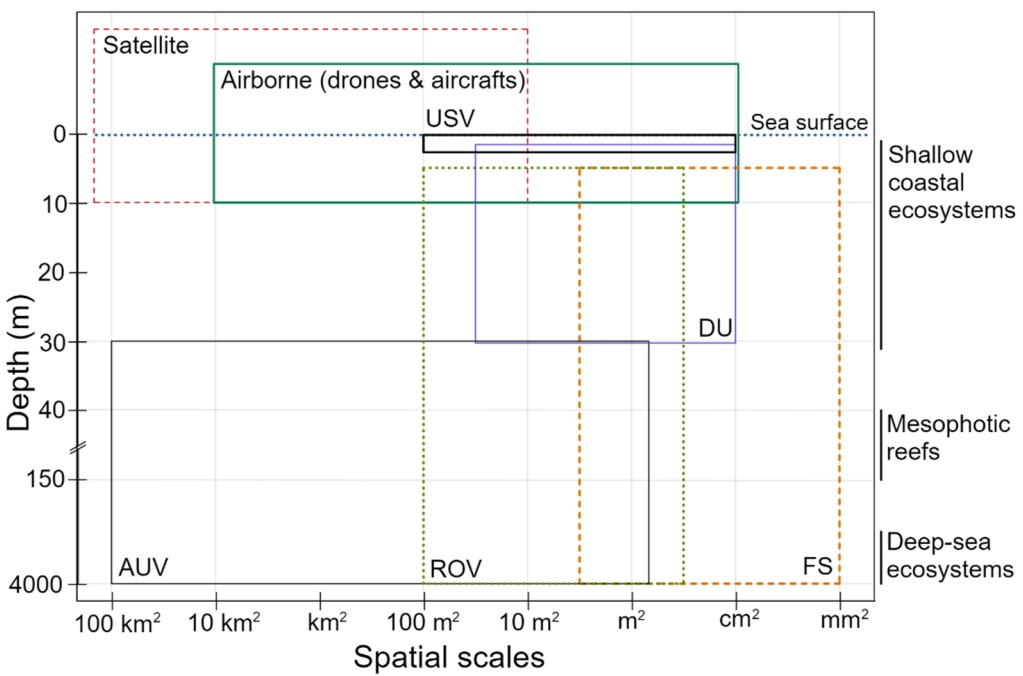

**Figure 1.** Present spatial scales of observation for hyperspectral imaging platforms in a marine context. This highlights the spatial extent to which each system can optimize its observations, plotted against the depth range of the system. Acronyms are defined by: unmanned surface vehicle (USV), diver units or systems (DU), autonomous underwater vehicle (AUV), remotely operated vehicle (ROV), and fixed stations (FS). Airborne includes drones and manned aircraft.

### 1.2. Marine Benthic Imaging—A Tale of Three Bands

Underwater platforms equipped with traditional RGB cameras for the acquisition of high spatial resolution (<1 cm) digital images of benthic organisms and environments have shown considerable advantages through being non-invasive, by reducing in-water survey time and providing a useful permanent archive of surveyed ecological data [31–33]. The continuous development of optical cameras and sensors mounted on underwater platforms capable of proximal sensing surveys of the seafloor includes towed cameras and unmanned vehicles comprising both remotely operated vehicles (ROVs) and autonomous underwater vehicles (AUVs). Close-range platforms are enabling small-scale (~cm) observations with co-located physical sample collection (e.g., specimens or sediment) on new environments such as the deep seafloor, as well as increasing the geographical extent of these surveys (m–km) [34]. An ROV or AUV survey can collect datasets of thousands of images from a single deployment [35,36]. Similarly, cameras fixed to the seafloor for studying seafloor processes of variable temporal scale (days to years, i.e., time-lapse studies) provide thousands of images and videos from one deployment [37].

However, advances in the data acquisition stage (i.e., number of images and dataset sizes) have surpassed the analysis capacities of human operators to translate the benthic images into ecological data ready for statistical analysis (i.e., image annotation) (Figure 2), causing a "bottleneck" in marine ecological research [38]. Estimates are that only 1–2% of image data is actually processed [39,40]. Furthermore, human annotators can introduce subjective errors into the analysis [36,41]. Standardized image annotation protocols (e.g., CATAMI) [42], tools (e.g., BiiGle) [43], machine learning algorithms [44,45], and marine object-based image analysis of photomosaics [46–48] are alleviating different problems within the image-analysis workflow (e.g., image annotation, different pixel size). However, the analysis of large image datasets still relies on manual methods because: (1) there are occasions where benthic heterogeneity and complex morphologies of underrepresented taxa demand human attention and cognition for image annotation [38] and (2) machine learning algorithms still require large training datasets of manual annotations to generate

accurate estimates [38,44]; both reasons restrict full automation in the analysis of benthic images.

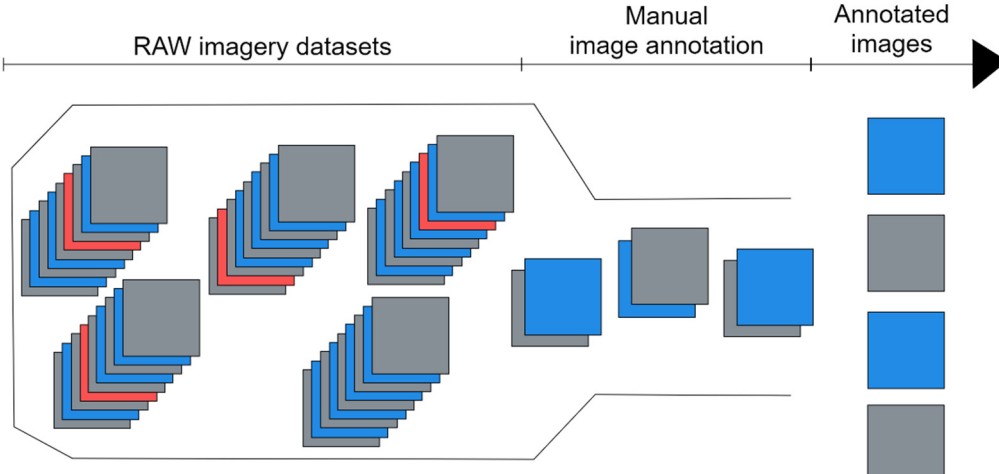

**Figure 2.** Seafloor image annotation bottleneck attributed to the large RGB image datasets and the limited automated capabilities.

Automating benthic image annotation is a complex challenge as accurate species identification often requires human experts to observe microscopic or contextual features. The taxa, feature, or process identification could be improved by increasing the image spectral resolution. For example, standard benthic imaging surveys typically rely on cameras with a high spatial resolution (~cm) but a low spectral dimension, as only three "spectral" bands are acquired per pixel (red, green, and blue, or RGB) with a specific sensitivity inherent to each camera model (Figure 3) [31]. RGB bands are often broad, comprising wavelength information over 60–100 nm wide in the visible region of the electromagnetic spectrum and are not radiometrically corrected [12]. As such, the lack of spectral detail per pixel demands manual annotations due to "colour" confusion between pixels [48]. Color correction of underwater images is currently under development through the advancement of image restoration and computer vision algorithms [29].

Using RGB images for seafloor ecological studies is also limited by the analyst's capacity to evaluate and process them as well as the capability to detect small-scale ecological processes including phototrophic activity, biogeochemical properties of sediments, and other sediment–water interface interactions [28,31]. We, therefore, require the ability to integrate multiple instruments and physical sampling approaches to correlate fine-scale benthic processes with surrounding environmental variables [28,49]. For instance, benthic RGB time-lapse cameras qualitatively illustrate the seasonal supply of phytodetritus in polar oceans, with massive pulses of fresh organic material arriving over a few days or weeks [50,51]. Yet, current RGB image-based observations are described as "varying from white to green," or "ranging from pale yellow-green to dark green" [52], which can be considered subjective, difficult to standardize, and incapable of providing information about the timing and biogeochemical composition of phytodetritus that determines its nutritional value and carbon burial [53–55]. Both the lack of automation capacity and subjectivity compels for integrating multiple spectral bands in seafloor ecosystem studies.

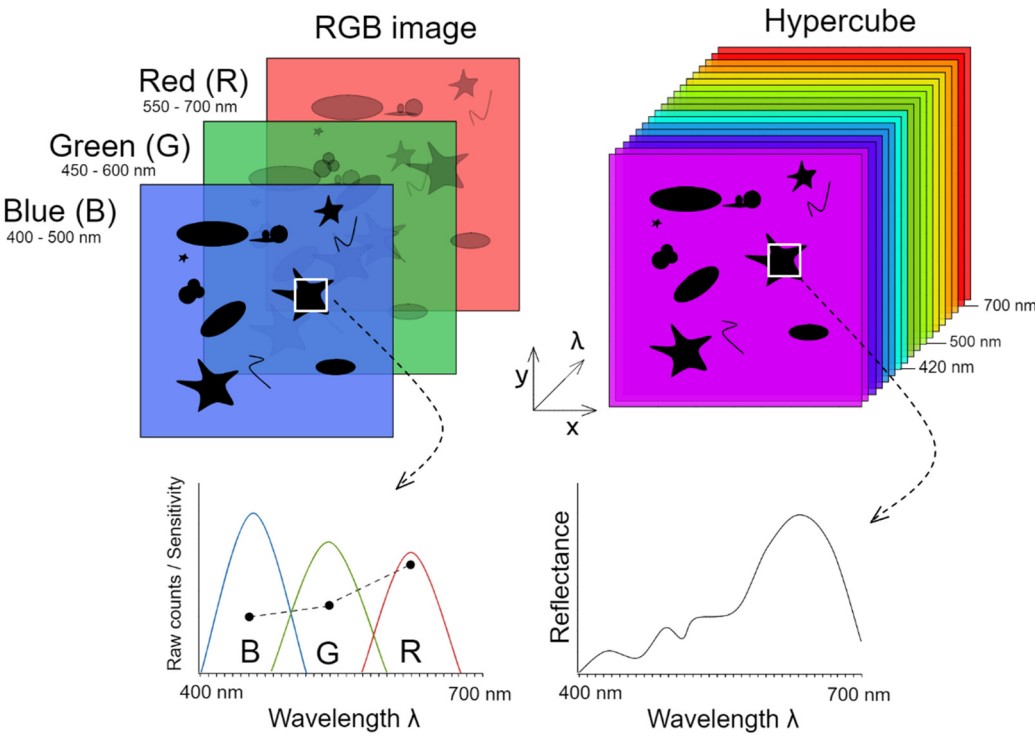

**Figure 3.** Comparison in the amount of information provided by an RGB image and a hyperspectral image cube from a heterogenous seafloor area.

## 2. Methods and Scope of This Review

Increasing advances in the ecological theory and observation capacity granted by hyperspectral imaging in terrestrial environments [56,57] compel us to examine the capacities and implications for seafloor ecosystem research. We review the main contributions and recent developments of UHI research for benthic organism quantification, identification, and mapping applications for marine ecosystem research [28,29,49]. Along the process, we discuss the platforms available for deploying UHI systems and the trade-offs between them for surveying different environments. We also highlight promising applications of UHI in automated benthic organism identification, ecosystem process studies, and an overview of the challenges for repeatable and accurate UHI data collection.

This literature review encompasses underwater applications of hyperspectral imaging from 2013 to November 2020, focusing on peer-reviewed journals in English. The search was performed with Google Scholar and PubMed using a combination of keywords, namely *underwater hyperspectral*, *spectroscopy*, *imaging*, *benthos*, *marine*, *mapping*, and *seafloor*. Several articles have used "hyperspectral imaging" for marine biological studies; however, we only included literature where the hyperspectral system was submerged in an aquatic medium studying spectral signatures of marine organisms in vitro and *situ*. Studies that used imaging spectroscopy without submerging the instrument but for similar research objectives were included for discussion purposes. We have excluded information and studies targeting underwater mineral spectral signatures [58,59], archeology [60], and aquaculture [61].

## 3. Analysis of Underwater Hyperspectral Imaging (UHI)

Compared to RGB cameras, or other multispectral sensors, where each pixel samples broad discrete digital values associated with portions of the electromagnetic spectrum (see Section 1.2), the "contiguous" spectral sampling is a key feature that differentiates hyperspectral imaging from other broad band sensors. A hyperspectral image consists of a

three-dimensional (x, y, λ) data cube where x and y represent the spatial dimension and λ the spectral dimension (Figure 3) [14,26].

Objects, or any surface of interest, absorb and scatter light at specific wavelengths (or frequencies) of the electromagnetic spectrum based on their molecular and structural properties, along with the directional components of the light source to the object [62]. Sampling hundreds of contiguous spectral bands enables researchers to distinguish objects and their attributes by their characteristic reflectance spectrum—also called the "spectral signature" or "optical fingerprint." The spectral resolution achieved by UHI (~1–15 nm) permits the retrieval of information about the biogeochemical composition of the seafloor [26,27,63], and the specific absorption wavelengths of natural pigments or other molecules found in marine organisms [64]. When enough spectral signatures of the same type of object are acquired, they can be grouped to build "spectral libraries." These in turn facilitate automated identification of benthic features and organisms through statistical learning techniques [64,65].

### 3.1. Applied UHI Systems and Sensor Architectures

Currently, seafloor UHI applications have only employed a sensor architecture referred to as a push-broom sensor, aptly named because it captures one line of pixels at a time and, through a straight-line movement of the mounting platform, renders a transect image [26]. Due to the considerable attenuation of light compared to above-surface applications, either due to scattering and absorption of particles or the water itself, a critical consideration is typically made to equip the system with appropriate light sources able to illuminate the seafloor to achieve an adequate signal-to-noise ratio. This "active" sensing approach also helps to evenly illuminate complex seafloor topographic heterogeneities, particularly when the sun is at an angle off the nadir direction. In other words, there must be enough light for it to be transmitted through the water medium, reach the seafloor, be reflected, and then refracted or dispersed into multiple wavelengths while still retaining a meaningful signal from the seafloor object [12,66]. The illumination source should emit light over the study spectrum (i.e., visible range) in a uniform manner. Ideally, the whole study area should be illuminated without shadows to acquire seafloor spectral signatures that provide reliable information about its biogeochemical composition. As such, an "active" approach with platforms carrying high-power light sources is usually preferred [26]. However, recent studies have shown it is possible to acquire high-quality underwater spectral signatures relying only on solar irradiance (i.e., a "passive" approach) as done in particular in environments such as sea-ice where imagery was acquired in the transmission mode [67,68].

The appropriate platform, sensor, and illumination required for UHI applications will depend on the research question, available resources, and the environment in which the survey will be conducted (see options in Figure 1). Miniaturization and automation of remote-sensing payloads are always preferable but are inevitably associated with increased cost and/or complexity [69]. Current UHI platforms include motorized rails, unmanned underwater vehicles (UUVs; ROVs and AUVs), unmanned surface vehicles (USV), diver-operated units, under-ice sliding units, and geo-stationary platforms (Table 1). Positioning instruments (e.g., ultra-short baseline) or inertial measurement units provide high-quality navigational data to permit the rows of hyperspectral pixels to be spatially referenced whilst retaining the geometric accuracy of the surveyed area [26,67]. The platform of choice will also be influenced by the spatial scale of the features to be mapped (cm–m), the resolution required for classification, and engineering specifics of the system (e.g., processing and storage capacity, payload power supply, etc.).

**Table 1.** Overview of current UHI systems for data acquisition along with their seafloor mapping capabilities and tradeoffs.

| System/Platform | Achieved Transect Length (m) | Possible Survey Area (~m²) per Deployment | Spatial Resolution Achieved (cm/pix) | Distance to Target (m) | Deployment Depth (m) | Operation Mode | Reference |
|---|---|---|---|---|---|---|---|
| Underwater rail | 1–5 | 10 | 0.1 | 1 | 5–20 | MO from surface, A capacity | [26,27] |
| AUV | Not defined | $1 \times 10^9$ | 0.6 | 8.5 | 2300 | A | [26,70] |
| ROV | 1–20 | <500 | 0.1 | 1 | 30–4000 | MO from boat | [71–73] |
| USV | 1–20 | <500 | 0.5 | 1.5 | Surface | A | [67] |
| Under-ice slider | 10–30 | <40 | 0.1 | 1.2 | 1.5 | MO above ice | [68] |
| Diver units | 50 | 500–650 | 0.4 | 1 | 30 | MO underwater | [74] |
| Fixed stations | 1 | <2 | 0.1 | 1 | ~3500 | MO from boat, A capacity | [75] |
| Lab systems | 0.01 to 1 | N/A | 0.05 | <1 | - | MO or A | [27,72,76,77] |

Autonomous underwater vehicle (AUV), remotely operated vehicle (ROV), unmanned surface vehicle (USV), manual operation (MO), autonomous (A).

### 3.1.1. Fixed Underwater Motorized Rails

Due to the complexity of acquiring UHI imagery using a push-broom sensor architecture, initial deployments of UHI systems began mostly as a "proof of concept" by being deployed on camera rails mobilized using small electric motors (example in Figure 4a). These platforms demonstrated that the images obtained could be useful in providing evidence of micro-scale processes on the seafloor (cm) [27], as well as habitat and organism identification over small areas of interest (<10 m survey line) [26,64]. The strength of these "stationary acquisition platforms" has been to test hypotheses before being "scaled up" to moving platforms [61,66,72,77]. Placing these electric rails at a single location on a set of stable tripods, for example, reduces any need for complicated platform motion tracking (x, y, z, pitch, roll, heading). It, therefore, minimizes the need for any geometric rectification and geolocation algorithms inherent to push-broom image acquisition and processing [78].

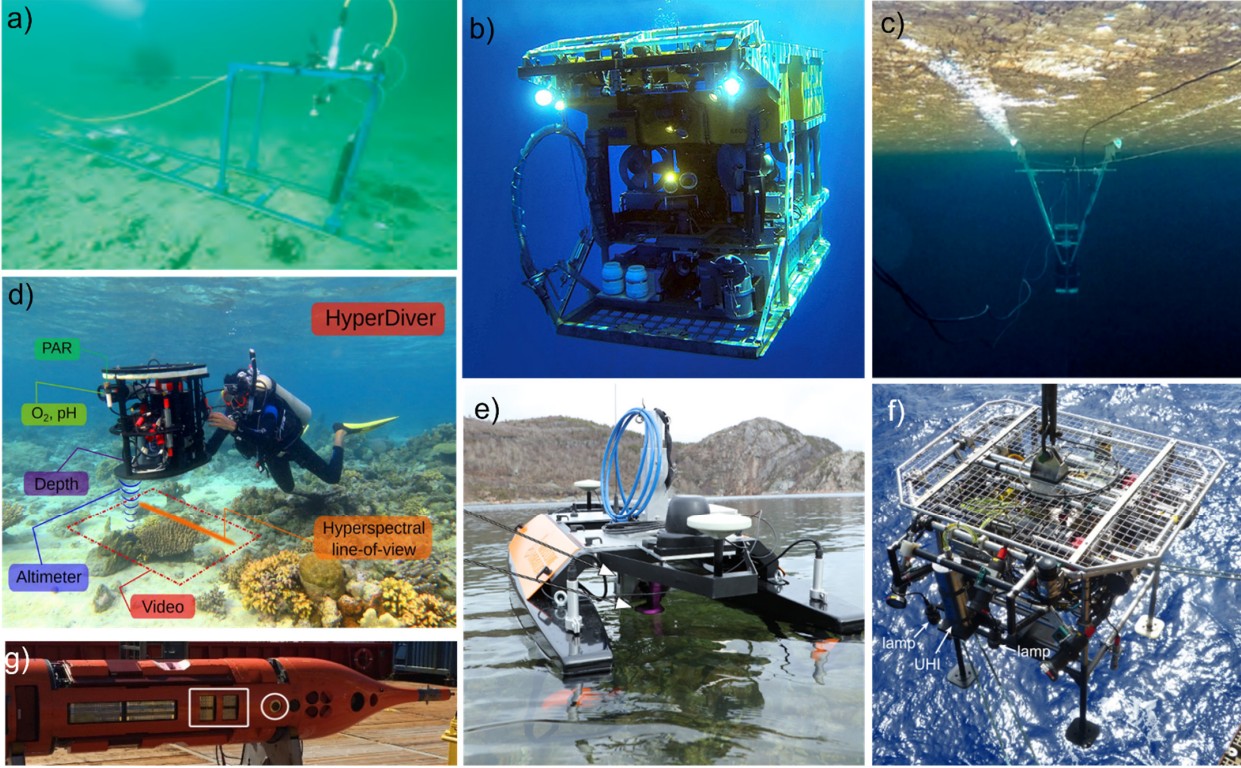

**Figure 4.** Photographs of platforms employed for in situ UHI studies. From (**a**–**g**), (**a**) electric rail, (**b**) remotely operated vehicle (ROV), (**c**) under-ice sled, (**d**) diver-operated unit (DU), (**e**) unmanned surface vehicle (USV), (**f**) fixed stationary platform (FS), and (**g**) autonomous underwater vehicle (AUV).

### 3.1.2. Underwater Unmanned Vehicles (UUVs)

UUVs, such as AUVs or ROVs, provide the means to efficiently acquire hyperspectral imagery at spatial scales ranging from 1 $m^2$ to 1000 $km^2$ [26] (Table 1). AUVs have been suggested as "the best platform for UHI mapping over large areas (1000 $km^2$) of seafloor" [26] (Figure 4g); however, their widespread adoption as an imaging platform for UHI is yet to be demonstrated. Few studies have used AUVs for hyperspectral sampling of the benthos [63,70,79]. Within the published literature, the most used platform for UHI studies of marine environments over smaller-scale areas are ROVs deployed from a ship (Figure 4b). These platforms are convenient as they have an unlimited power supply for energy-demanding onboard light sources and are easier to operate compared to larger AUVs [63]. UHI systems mounted on ROVs have proven useful for acquiring deep-sea benthic data (~4000 m) and for automated organism detection and classification [71]. ROV employment, advantages, and requirements for quality UHI measurements have been discussed in detail by [26,63].

### 3.1.3. Unmanned Surface Vehicles (USVs)

USVs represent an advantageous platform for collecting UHI data in very shallow water environments that are difficult to access by other platforms with a larger draft (e.g., boats) (Figure 4e) or that have not been previously navigated so survey lines cannot be pre-planned (e.g., as required for an AUV) [67]. These platforms can theoretically map areas of a similar extent as ROVs (Table 1). Advantages of navigational data provided by above-water GPSs and a USV as an imaging platform presented by [67] provided evidence of how to map a shallow marine habitat with overlapping hyperspectral imaging transects. Despite complex water optical properties in near-coastal regions (e.g., chlorophyll *a*, dissolved organic matter), [67] explains that UHI data comprises enough signal for accurate benthic organism classification.

### 3.1.4. Under-ice Sliding Platforms

Tailored systems for UHI can and have been developed for monitoring marine habitats that require customized solutions. For example, [68] demonstrated an inverted under-ice sliding platform for surveying photosynthetic sympagic microalgae beneath land-fast sea-ice in Antarctica (Figure 4 and Table 1). In this study, the under-ice surface was relatively flat, allowing deployment of a slider and "skiing" platform to retrieve straight UHI transects without the complexities of positioning corrections and georectification algorithms (Figure 4c). An important difference of this system from other UHI applications is that light is being captured in transmittance mode (i.e., solar irradiance passing through the ice and interacting with microalgae and the water column before reaching the sensor). [68] discusses that adding a high-spatial-resolution RGB camera maximizes the information collected by the platform, as it enables the reconstruction of the under-ice surface via *Structure-from-Motion* photogrammetry, which could eventually support push-broom imagery georectification [68,78].

### 3.1.5. Diver Operated Units (DU)

Chennu et al. [74] demonstrated the capabilities of a diver-operated unit to collect underwater hyperspectral data on shallow coral reefs (Figure 4d). This platform could detect individual spectra of mixed benthic assemblages that were otherwise difficult to discriminate from aerial or satellite images. Furthermore, the integration of different instruments on the platform (e.g., bathymetry, pH, $O^2$) maximizes the information gathered during the survey by one person. Yet, [74] explains that diver-operated UHI systems are "large and require miniaturization to be integrated into autonomous vehicles," as well as technical training to operate. Compared to other platforms, diver units can easily be integrated into standard diver-based surveys and can bridge the link between field ecologists and the remote-sensing community [74].

### 3.1.6. Fixed Stations & Networks

Initial steps towards deep-sea UHI stations by [75] state that these are mainly suitable for small area studies (seafloor coverage of a few m$^2$) and that a flat seafloor for the safe landing of the platform is key (Figure 4f). Once landed it acquires measurements free from variations in altitude, pitch, roll, and heading that are inevitable from a moving vehicle [71]. Further, the biggest premise of deploying UHI on stationary platforms may become popular for underwater observatory networks [29]. Networks of stationary observatories in remote and deep benthic ecosystems could push UHI research to be integrated into protocols for autonomous data acquisition, automated processing for real-time analysis, storage, and access via internet connections [29,49].

### 3.1.7. In Vitro and Ex Situ-Based Systems

Electric rails from which the imaging system can be mounted (Section 3.1.1.) are suitable for analyzing samples extracted from the natural environment (*ex-situ*) or grown in an artificial medium (*in vitro*), permitting the calibration and validation of UHI observations [27,66], or for deployment in field laboratories in remote regions (e.g., polar study sites or at sea where working indoors in a lab is preferential) [76]. Laboratory setups for UHI systems also grant efficient temporal quantitative analysis of seafloor processes and their interactions under specific light intensities and spectral qualities (e.g., photosynthetic activity) [80] and capture dimensions that are not visible from the in situ surface perspective. For example, [27] and [76] have demonstrated the vertical variability found in natural and artificial mediums (e.g., soft substrates, sea-ice), and its influence on UHI analysis and interpretation. Furthermore, *ex-situ* systems allow the acquisition of spectral information of living organisms for them to be taxonomically identified [64], describe their pigment composition [64,72], or assess individual physiological responses [77].

## 4. Breakdown of Applications and the Importance of Pigments for UHI

Light (electromagnetic energy) availability and its spectral quality represent important elements to consider in our understanding of how current marine ecosystems have evolved, respond to, and contain relevant information that we can retrieve with UHI [62]. Ecological diversification can be driven by features that, for example, favor abiotic resource acquisition [81]. As an abiotic resource, sunlight reaching the seafloor is extremely variable over space and time [82]. Multiple factors have determined light availability to marine ecosystems through evolutionary time, diversifying the strategies of energy acquisition and exploitation [83–85]. In this review, beyond changes in the diel cycle and atmospheric conditions (e.g., clouds or winds influencing surface properties), we consider three main components that determine light availability to seafloor ecosystems. First, the water body itself absorbs and attenuates wavelengths of light increasingly towards the reds (>550 nm). In clear ocean waters, red light disappears at a depth of 15 m; as depth increases, only blue wavelengths remain [62,86]. Secondly, colored dissolved organic matter contains diverse absorption features mainly captured by the yellow wavelengths (~575 nm) [87]. Third, phytoplankton concentration attenuates the photosynthetic active radiation (400–700 nm) available to the benthos through light-absorbing pigments [26].

Once light reaches the seafloor, pigmentation of marine organisms plays a central role in defining the spectral signature acquired with any UHI sensor. In biology, a pigment (or biochrome) is any molecule that, through selective absorption of light, results in the representation of colour in micro- and macro-organisms. Pigments occur in virtually every taxonomic group, from bacteria to animals. Green algae (Chlorophyta and Charophyta), for example, utilize chlorophyll (Chl) *a* and *b* together with different carotenoids that sharply absorb red and blue wavelengths for photosynthesis [83]. The least absorbed and most reflected wavelengths are the green ones that are transmitted to the sensor. Other marine photosynthetic organisms use other pigments that absorb other wavelengths, such as Chl *c*, fucoxanthin, and phycobilins [85]. On the other hand, pigments in marine heterotrophic organisms (e.g., invertebrates) serve different purposes such as mimicry, advertisement, or

warning, to name a few [88]. In shallow marine environments that are exposed to solar radiation, bright organism colouration is widespread (e.g., coral reefs), even for organisms in darker areas where colours are only visible to humans with artificial illumination. For instance, compared to the human eye that is sensitive to three colors (i.e., RGB), marine organisms such as the mantis shrimp possess cells sensitive to 13 wavelengths covering from the UV range to the reds [89]. Alternatively, certain polar photosynthetic organisms present unique pigmentation and photophysiological strategies that allow them to survive seasonal darkness or rapid changes in under-ice irradiance [90–92].

In the following sections, we summarize and explain the role of pigments in UHI data analysis and its applications for seafloor ecosystem studies (Figure 5). UHI data analysis can be grouped into two categories: (1) classification on a per-pixel basis of discrete features (e.g., benthic habitat mapping) and (2) regression and prediction of a biogeochemical feature (e.g., photosynthetic pigments) on a per-pixel basis (Table 2). Classification algorithms can be supervised or unsupervised [14]. Supervised algorithms make maps using input variables (e.g., spectral signatures), which are then translated into categorical features useful for assessing benthic cover, organism abundance, or physiological status [93]. Regression algorithms estimate a mapping function based on a feature from input variables (e.g., wavelength absorption, spectral indices) to produce an output variable (e.g., Chl *a* content, a continuous variable) [94]. Applications of hyperspectral imaging for marine environments and organisms are still in the early stages of development; here, we report the findings of all studies in our knowledge as they all have something to add to this new field. Further, we address the type of data analysis used for each environment or taxa, trade-offs between the type of analysis performed, and the challenges of UHI for obtaining replicable and verifiable pigment-specific signatures for seafloor ecosystem studies (Table 2).

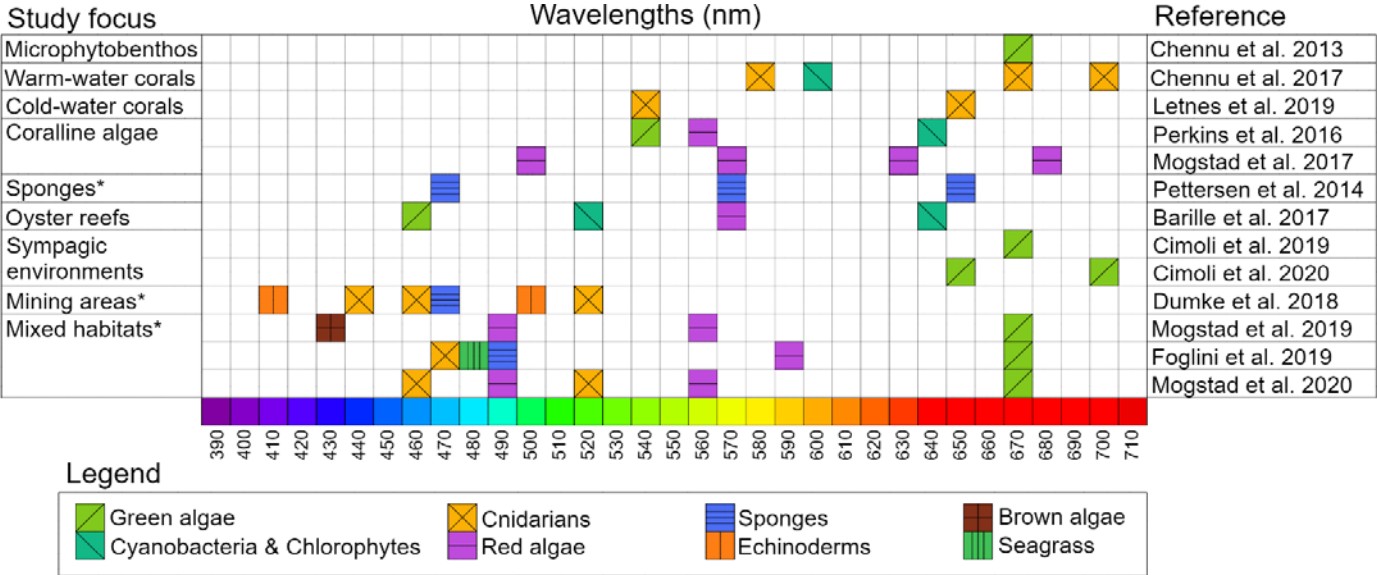

**Figure 5.** A summary of studies using UHI focusing on marine environments or organisms (left column). The bands shown on this figure have been employed for organism detection, or classification*, or estimation of pigment abundance. The different biota is indicated by different colours/patterns. Wavelengths were binned down to 10 nm intervals ranging from 390 to 710 nm. Presentation of binned wavelengths of pigment absorption was adapted from [57].

### 4.1. Microphytobenthos and Sediment Phytodetritus

Microphytobenthos (MPB) are unicellular eukaryotic algae (e.g., diatoms) and cyanobacteria that inhabit the top few millimeters of shallow sediments (i.e., the sediment–water interface) [53,95]. In some cases, the primary production of these organisms can be greater than that of phytoplankton in the water column, providing an important food source for benthic organisms [95]. The spatio-temporal variability of microphytobenthos

is influenced by multiple abiotic and biotic factors that require small-scale observations (~cm). The need for non-destructive standardized methods that provide information on the abundance and distribution of phytobenthic cells has led to the development of UHI methods to detect and quantify Chl *a* (Table 2) [27,80].

*In situ* UHI research by [27] focused on the small-scale (1 m$^2$) temporal variability of primary productivity caused by MPB organisms in intertidal sediments. Tailored spectral indexes based on Chl *a* absorption features were used to quantify daily differences of in situ Chl concentrations at a sub-millimeter scale. These observations revealed the role of polychaetes (annelid worms) in enhancing primary productivity at small spatial scales (~cm). For example, sediments bioturbated by *Arenicola marina* displayed a heterogeneous MPB distribution as to where: the polychaete feeds and removes most MPB cells from the sediment and its interspersed, while Chl *a* concentrations are elevated between mounds where grazing has not occurred [27].

On the other hand, *ex-situ* close-range hyperspectral imaging of subtidal MPB has also illustrated how the invasive gastropod species *Crepidula fornicata*, now widespread along European shallow coasts, enriches the sediment at small spatial scales (1 cm$^2$) with organic excretions and by modifying micro-hydrodynamics [80].

The non-invasive nature of close-range hyperspectral imaging demonstrated by [27] and [80] can lead to an improved understanding of microphytobenthic ecology, including regulation mechanisms, such as bioturbation, grazing, nutrient enrichment, and circadian cycles with proper sediment validation samples.

### 4.2. Coral Reefs

Warm- and cold-water corals are known to create reefs that provide a critical habitat for thousands of species in shallow and deep environments. Despite the benefits provided to ocean ecosystems, these environments are being damaged by human activities both at local and global scales [21]. The diversity of species found within small spatial scales in coral reefs requires tools able to cover the vast extent of these environments without compromising the observation of small-scale features. Ongoing UHI research has shown the potential to overcome the challenges of surveying both warm- and cold-water coral habitats.

#### 4.2.1. Warm-Water Corals

Warm-water coral reefs are dominated by scleractinian (stony) corals that rely on their symbiotic relationship with zooxanthellae, a group of photosynthetic dinoflagellate protists from the genus *Symbiodinium* [96]. Coral spectral signatures are determined by differences in zooxanthellae pigment composition and concentration, which show consistent spectral signatures across biogeographic regions [97]. Furthermore, overall health (e.g., bleaching) can be spectrally determined by the loss of pigmentation of the host [98,99].

In their research using a diver-operated UHI system, [74] demonstrated that the spectral resolution of UHI permits 1) the production of benthic habitat maps with accurate species identification and 2) estimates of photosynthetic activity at small spatial scales (e.g., 1 cm$^2$). Both observations are generated by specific wavelength absorption (Figure 5). For example, the second derivative of the wavelength of maximum absorption of in vivo chlorophyll pigments (670 nm) was used for calculating the concentration of Chl at each pixel. Similarly, the first derivative of the infrared shoulder of Chl absorption (700 nm) was used to discriminate coral from sand and algae. Furthermore, the second derivative at 580 nm was found to be consistent over certain species of stony corals. Finally, the second derivative at 605 nm, the absorption peak of phycoerythrin (accessory pigment of cyanobacteria and chryophytes), was found to be consistent with sediment regions. [74] explains that although these spectrometric values provided good results, the quantitative aspect of this research remains to be validated and calibrated (Table 2).

*In vitro* proximal hyperspectral imaging has emerged as a tool to evaluate coral health (e.g., overgrowth of algae or bleaching) [99,100]. Early research by [100] combined spectral

analysis with dissolved oxygen measurements to understand interactions between corals and algae. While hyperspectral imagery successfully distinguished different photosynthetic organisms through their characteristic spectral signatures, oxygen profiles identified the type of competitive interaction between benthic reef organisms (e.g., fleshy algae create hypoxic zones detrimental for coral survival) [100]. Recently, [101] evaluated in vitro the fluorescence emission spectra of several warm-water coral species to quantify the spectral signal of coral bleaching. Samples were exposed to increasing temperatures whilst being imaged with a hyperspectral camera every 24 h. Through this process [101] were able to spectrally detect the expulsion of the algal symbiont (Table 2). Although [101] explains that underwater platforms, like ROVs, would require powerful ultra-violet and blue light sources to excite fluorescence emission of corals from the surveying altitude (approx. 1 m). However, this demonstrates the potential of UHI for automatic and rapid physiological assessments of coral reef health.

### 4.2.2. Cold-Water Corals

Cold-water stony corals also create reefs but at depths where they rely solely on particle feeding, rather than photosynthetic symbionts. *Lophelia pertusa* is the most abundant deep-water coral and is known for forming deep-water reefs that are considered biodiversity hotspots [22]. The role of underwater digital imaging systems has been critical in revealing their widespread distribution, their ecological role in providing habitat to many species in deep ocean environments, and the extent of human impacts to these ecosystems [22,102]. As such, recent research has demonstrated how ROVs represents an essential platform for UHI surveys of cold-water coral habitats to evaluate incremental anthropogenic disturbances to these habitats [71,77].

In their in vitro UHI research, [77] explained how oil spills and mining represent a direct threat to these habitats and explored the utility of UHI classification methods for detecting *L. pertusa* physiological status after weeks of pollutant exposure in experimental tanks (Figure 5). To achieve this, [77] evaluated the reflectance spectra of coral samples exposed to different concentrations of a toxic hydrocarbon. The successful classification of coral polyp health and mortality based on their reflectance spectral signature demonstrates the potential for spectral signatures to be used to monitor the physiology of different benthic organisms (Table 2). Furthermore, [77] claim this work is the first step towards non-invasive automated methods for in situ mapping of cold-water coral physiological conditions. However, further studies should involve more species and pollutants to consider them robust or operational as a method.

An upshot of recent UHI research in cold-water coral reefs by [73] is that if significant reference spectra have been catalogued, spectral libraries can become a quick and reliable benthic habitat classification tool (Table 2).

### *4.3. Coralline Algae*

Coralline algae are a cosmopolitan group of calcifying red algae (Rhodophyta, Corallinaceae), acting as ecosystem engineers in almost every coastal ecosystem [103]. A growing concern about climate change impacts on coralline algae (e.g., ocean acidification) has led to increased efforts to determine their functional roles [104] and distributions [105,106]. Their characteristic colouration attributed to pigments that absorb in the blue-green wavelengths (Figure 5), along with the challenges associated with determining their key functional roles for coastal ecosystems have led to hyperspectral imaging applications as a bio-optical tool for crustose coralline species classification (Section 4.3.1) and to reveal millimeter-scale variability of branching coralline communities (Section 4.3.2) (Table 2).

### 4.3.1. Non-Geniculate (crustose) Coralline Algae

Foglini et al. [72] demonstrate UHI as a developing technology for identifying and estimating crustose coralline algae (CCA) abundance based on their spectral signature. Despite the low sample number (n = 4), the authors explain that the photosynthetic

accessory pigment *R*-phycoerythrin (*R*-PE) makes CCA a spectrally conspicuous group that can be distinguished in their natural habitat with UHI. In the investigation in [72], all four species displayed similar spectral signatures, with *R*-PE and Chl *a* representing the primary light absorption components. However, [72] found differences in spectral intensity between species. In other words, all CCA species assessed absorbed light similarly, resulting in similar spectral signatures; yet, there were consistent differences in the amount of light absorbed by each coralline taxa, possibly caused by the amount of *R*-PE per specimen. Interestingly, the in situ classification accuracy of CCA versus non-coralline substrates was enhanced when considering the average spectral signature of different CCA species, rather than the individual species signature (Table 2).

The study by [72] draws attention to the fact that the green wavelengths (500–565 nm) provided most of the variability in the principal component analysis. This coincides with the absorbance spectrum of *R*-PE, resulting in the conclusion that "coralline algal *R*-PE content could serve to separate species spectrally" (Figure 5). Additionally, other descriptors of CCA morphotypes, like tissue/crust thickness or pigment packaging effects should be explored with UHI to provide functional approaches to this benthic group [107].

### 4.3.2. Geniculate Coralline Algae

Branching (i.e., geniculate) coralline algae serves as an important habitat to multiple macro- and micro-epiphytes. In vitro hyperspectral imaging has shown unique capacities to reveal micro-spatial patterns (~1 mm$^2$) of these complex communities. [108] demonstrated that specimens from different light environments (lower and upper shore) host different communities of epiphytes and that even within an algal specimen, epiphyte densities compositions vary. By performing double derivative analysis of each wavelength corresponding to specific absorption wavelengths of diatoms (546 nm) and Chlorophyta (648 nm), [108] found a spatial pattern in their distributions, with abundances increasing from the base to the apex of the frond (Figure 5). As such, they found a decrease in the inherent red algae pigment of *C. officinalis* (phycoerythrin, 568 nm) along the frond caused by epiphyte cover (Figure 5). Their results demonstrate the effectiveness of UHI for the detection of photosynthetic microbiome pigments. However, the presence of epiphytic communities associated with macroalgae may influence overall photosynthetic activity estimates. Further studies are required to examine the influence of epiphyte cover on hyperspectral imaging estimates of coralline algae productivity.

### *4.4. Sponges*

Marine sponges (Porifera) occur at all water depths. Still, uncertain effects cause vast sponge aggregations composed of a single species or mixed assemblages. Aggregations can extend hundreds of km$^2$, increasing the three-dimensional structure of the benthos, modifying the small-scale hydrodynamics of the sediment-boundary layer, and enabling a myriad of organisms to inhabit sponge fields [109,110]. Sponge habitats are vulnerable to persistent disturbances such as oil drilling, long-line, and trawl fishing activities, which have the potential to discharge sediments and chemicals, and/or destroy the habitat. Yet, we know little about sponge-dominated systems as they are understudied, resulting in many regions lacking sufficient information for determining their protection status [109].

The relationship between the spectral signature and the pigment composition of sponges was evaluated by [64] to use UHI as a bio-optical taxonomic tool (Table 2). Their study focused on using different pigment extraction methods to (1) identify unknown pigments, (2) determine the organism pigment composition, and (3) relate the UHI spectral signature to the different pigments. The results of [64] showed that UHI is a potentially powerful benthic identification tool only with *a priori* knowledge of the pigment composition of the organism. They noted that for accurate classifications, care should be taken as "certain taxa may have diverse colourations and optical signatures across geographical or ecological areas," meaning that standardising a spectral signature for one species may be difficult if not impossible.

Pettersen et al. [64] found five pigments in the sponge *Isodictya palmata*, which is an interesting observation as sponge pigments come from different sources and may serve or reflect different biological purposes (Figure 5). For example, sponges are known to host symbiotic organisms that can endure both the digestive and immune processes of their sponge host [88]. Some species of bacteria are pigmented and able to synthesize carotenoids, which absorb in the visible spectrum and contribute to the colour of the organism [111]. By performing pigment extraction along with UHI analysis, [64] found that derivatives from the pigment 2,6 benzathiazolediol, produced by the bacteria *Micrococcus* sp., were present in *I. palmata*. Another pigment found in this sponge was erinacean, which is known to be an antibiotic and a cytotoxic substance in the Antarctic sponge *I. erinacean*. In addition, two other pigments were detected, Calicogorgin B, which is also documented to have anti-predatory activity, and Aspergamide B, which is produced by marine fungus and absorbed through filter feeding.

Recent UHI studies by [71] showed that spectral signatures proved useful to discern whether sponge individuals of different morphologies belong to the same species; however, their identifications were based on RGB video inspection, not on pigment extraction validations. Furthermore, recent UHI studies by [73] disclosed that different sponge species have similar spectral signatures to each other, even those with different morphotypes; thus, it was not possible to determine a species solely based on the reflectance spectral values captured by the UHI. Therefore, [73] used previous publications to infer the most likely species presence and kept four different sponge classifications. As such, [64] highlights that knowledge about host/epibiont distribution must be considered when comparing UHI spectral libraries from multiple study sites with different environmental conditions. In our view, UHI has the potential to acquire additional information on sponge trophic ecology from a non-destructive survey sampling method; yet, extensive sampling or in vitro studies are required to further test host/epibiont spectral signatures.

### 4.5. Oyster Reefs

Aggregations of oysters on coastal areas form three-dimensional structures that serve as habitats to multiple species and perform important biogeochemical processes, such as biofiltration and nutrient recycling, which are of social and economic importance [112]. Their shell morphology and filtering activity modify local hydrodynamics, and they enrich the sediment through their excretions, which in turn are available to primary producers. Like other ecosystem engineers (see Section 4.3.2), oyster shells host photosynthetic micro-epibionts on their shells, the diversity of which is spatially variable, that may contribute to local-scale primary productivity, which is compelling for scalable methods able to characterize the heterogeneity found in these environments.

Barillé et al. [113] illustrated through the use of spectral index analysis (normalized difference vegetation index) that all oyster shells host photosynthetic micro epibionts (Table 2). In [113], hyperspectral imaging enabled identification of the epibiont composition via a derivative analysis of reflectance of the indicative absorption wavelengths of photosynthetic pigments of diatoms (462 nm), cyanobacteria (524 nm), rhodophytes (571 nm), and chlorophytes (647 nm) (Figure 5). Like other photosynthetic microbiomes (see Sections 4.1 and 4.3.2), UHI holds potential for revealing the complexity of oysters enhancing local productivity.

### 4.6. Sympagic Environments

Environmental change is modifying sea-ice physical properties (e.g., ice thickness and snow depth) concomitantly with its biological properties (e.g., biomass and photophysiology of its associated sympagic communities). Changes in sea-ice biophysical properties are expected to have cascading effects on polar marine food webs, primary productivity, and biogeochemical cycling [114]. Sea-ice supports diverse and often abundant communities of primary producers and consumers that sustain in part these ecological functions. Sea-ice algae are a key food source for higher trophic levels and display vertical and horizontal

variations that range from the meso- to the millimeter-scale, fluctuating on a daily, weekly, monthly, and seasonal basis [114]. UHI payloads are expected to fill a niche gap in mapping fine-scale sea-ice biophysical properties in a non-invasive manner at sub-mm spatial resolutions.

Cimoli et al. [68] first retrieved in situ proxies of sea-ice algae biomass on a sub-mm/pixel spatial resolution over 20 m long transects. This study showed that sea-ice algae biomass measurements can be made without destructive core sampling, which is also limited by being point-based and labour-intensive. The authors used an "inverted" ice sled as a platform for a coupled UHI-RGB system to improve the resolution for sea-ice algae biomass monitoring through (1) spectral indices as proxies of biomass and (2) under-ice topography of fast-ice environments in Antarctica (Table 2).

Further, Cimoli et al. [68] adds that the system capabilities and the "inverted" approach could provide information on more biological features than first thought, such as photo physiology, algae species composition, and habitat features of under-ice grazers. Recently, [76] discussed how UHI approaches combining both in vitro and in situ studies hold great potential for quantitive mapping of sea-ice algae variability, especially when the ice matrix is evaluated both vertically and horizontally to untangle the complexities of sea-ice primary productivity.

### 4.7. Seafloor Areas with Mineral Resources

Marine mineral exploration has moved from shallow coasts to off-shore deep seafloor environments [115]. The lack of knowledge about natural processes occurring in deep-sea ecosystems [22], and the impacts that current mining technologies could have on them, make deep-sea mining without biodiversity loss a challenging goal threatening the functioning of marine ecosystems [116]. The mining of resources found in benthic ecosystems can induce a complete removal, burial, or alteration of such environments in ways that, 26 years after a disturbance event, the physical effects are still lingering, impeding benthic communities to recover [117]. With an increase in seafloor areas being approved for mineral exploration, the development of methods that retrieve benthic community information with low bias in mining surveys (e.g., RGB image annotation by humans) is crucial for appropriate regional and international seafloor management [28,118].

**Table 2.** Underwater hyperspectral imaging (UHI) studies of marine benthic habitats and organisms. Validation refers to the technique employed to authenticate UHI imagery to other standard methodologies. Calibration refers to the technique employed to derive a classification or regression that can be applied to UHI imagery.

| Study Focus(Marine Organism, Environment) | UHI Application | Platform | Wavelengths/Resolution | Data Analysis Method | Validation | Calibration | Ref |
|---|---|---|---|---|---|---|---|
| Microphytobenthos | Spectral Index for Chl *a* | Electric rail | 400–900 nm @ 1 nm | Regression: spectral index | Pigment extraction | Chl *a* spectrophotometry | [27] |
| | Spectral Index for phycoerythrin | Diver-unit | 400–900 nm @ 1.5 nm | Regression: $\delta\delta$ (605 nm) | Visual ROI annotation | Uncalibrated | [74] |
| | Photosynthetic cell biomass | Electric rail | 400–1000 nm @ 1.3 nm | Regression: 3$^{rd}$ end-member spectrum | Visual ROI annotation | Uncalibrated | [80] |
| Warm water corals | Benthic classification | Diver-unit | 400–900 nm @ 1.5 nm | Regression: $\delta\delta$ (580, 675 nm) | Visual RGB | Uncalibrated | [74] |
| | Colony bleaching assessment | *In vitro* rotational stage | 400–1000 nm @ 2.2 nm | Object fluorescence emission spectra | Visual RGB | Spectrometer | [101] |
| | Physiological interactions | *In vitro* electric rail | 450–900 nm @ not specified | GOC$_{515\text{-}575\text{-}685}$ CIR$_{550\text{-}650\text{-}860}$ NDVI$_{800\text{-}680}$ ARVI$_{800\text{-}680\text{-}450}$ | Visual RGB and oxygen profiling | Uncalibrated | [100] |
| Cold water corals | Polyp mortality classification | *In vitro* electric rail | 381–846 nm @ 1 nm | Classification: *v*-SVM | Visual RGB | Visual inspection | [77] |
| | Benthic classification | ROV | 380–800 nm @ 15 nm | Classification: SAM | Visual RGB | N/A | [73] |

**Table 2.** *Cont.*

| Study Focus(Marine Organism, Environment) | UHI Application | Platform | Wavelengths/Resolution | Data Analysis Method | Validation | Calibration | Ref |
|---|---|---|---|---|---|---|---|
| Coralline algae | Bio-optical taxonomic tool | *In vitro* electric rail | 400–700 nm @ 2 nm | Classification: SAM | Pigment extraction | Spectrophotometry and HPLC | [72] |
| | Benthic classification | ROV | 400–700 nm @ 2 nm | Classification: SAM, MD, BE, SID, Pp | Visual ROI annotation | Spectrophotometry and HPLC | [72] |
| | Benthic classification | ROV | 380–800 nm @ 15 nm | Classification: SAM | Visual ROI annotation | N/A | [73] |
| | Classification of photo-epibionts | *In vitro* tripod system | 400–1000 nm @ 4.5 nm | Regression: $\delta\delta$ (546, 568, 648, 677 nm) | Pigment extraction | HPLC | [108] |
| Sponges | Bio-optical taxonomic tool | *In vitro* electric rail | 420–680 nm @ 1 nm | Classification: object reflectance spectra | Pigment extraction | HPLC & mass spectrophotometry | [64] |
| | Benthic classification | ROV | 378–805 nm @ 4 nm | Classification: SVM | Visual ROI annotation | N/A | [71] |
| | Benthic classification | ROV | 380–800 nm @ 15 nm | Classification: SAM | Visual ROI annotation | N/A | [73] |
| Oyster reefs | Classification of photo-bionts | *In vitro* electric rail | 400–950 nm @ 4.5 nm | Regression: $NDVI_{750-673}$, $\delta\delta$ (462, 524, 571, 647 nm) | Pigment extraction | HPLC | [113] |
| Sympagic environments | Proxy of ice-algae biomass distribution | *In vitro* electric rail | 400–1000 nm @ 1.7, 3.4, 6.8 nm | PCA | Pigment extraction | Uncalibrated | [66] |
| | Proxies of ice-algae biomass distribution | *In situ* under-ice sled | 400–1000 nm @ 3.5 nm | PCA $NDI_{441-426}$, $NDI_{648-567}$, $ANMB_{650-700}$ | Visual ROI verification | Uncalibrated | [68] |
| | Quantitative estimates of biomass via spectral indices for Chl *a* | *In situ* under-ice sled and ex situ electric rail | 400–1000 nm @ 1.7 nm | Regression: NDI, $AUC_{650-700}$, $ANCB_{650-700}$, $ANMB_{650-700}$, $LAUC_{650-700}$ | Pigment extraction | Fluorometer | [76] |
| Mineral resource assessment areas | Bio-optical taxonomic tool | ROV | 400–710 nm @ 4 nm | Classification: SVM | Visual ROI annotation | N/A | [71] |
| | Sediment deposition homogeneity | ROV | 400–700 nm @ 5 nm | Regression: PCA and singular-value decomposition | van-Veen grab | N/A | [119] |
| | Benthic classification | Stationary platform | 400–730 nm @ 2 nm | Classification: SVM | Visual ROI annotation | N/A | [75] |

Remotely operated vehicle (ROV), second derivative ($\delta\delta$), green-orange-chlorophyll (GOC), color infrared (CIR), normalized difference vegetation index (NDVI), atmospherically resistant vegetative index (ARVI), spectral angle mapper (SAM), binary encoding (BE), spectral information divergence (SID), minimum distance (MD), support vector machine (SVM), parallelepiped (Pp), normalized difference index (NDI), area under the curve (AUC), area under curve normalized to a maximal band (ANMB), logarithm-transformed area under the curve (LAUC).

In this context, UHI is proving to be a promising avenue for evaluating areas where increased levels of suspended sediments caused by exploratory drilling increase the mortality of filter-feeding fauna (e.g., corals and sponges), with drill-cause plumes extending up to kilometers away from a drilling site. [119] explains that current exploratory drilling environmental impacts are visually assessed with video-transects and manual annotators that classify the effects ranging from: (Class A) drilling site with smothered sediment, clear signs of recent sediment deposition, and absence of biological activity, to (Class B) undisturbed sediment representing natural conditions, no sediment deposition, and a diversity of organisms. The subtle differences between these two classifications (from A to B) are difficult to assess visually since natural and drill-cutting sediments can be similar in colour [119]. Although visual assessment and UHI results exhibit similar overall trends, the first UHI results show potential for reducing bias and automating the process (Table 2) [119]. In fact, [119] demonstrates decreases in the overall spectral similarity along a UHI transect as a function of the distance from the drilling location. A high correlation in the spectral similarity between sediment samples represents a homogenous sediment composition (class A), and, further away from the disturbed site, pixels exhibit more het-

erogeneity, reflecting natural higher complexity habitats with diverse fauna (and their corresponding spectral signatures; class B).

Further deep-sea research carried on by [71] demonstrated how classification algorithms using UHI data acquired in a deep-sea mining area provided far more detection of benthic organisms than visual identifications from RGB videos. For example, only three coral specimens were detected by human observers using videos, whereas the spectral classification detected 39 individuals. Furthermore, the high spatial resolution obtained by UHI represents an enormous benefit for the detection of smaller fauna and early-recruitment stages of sponges or corals (<2 cm in size), which are difficult to distinguish using standard RGB imagery (Figure 3). However, marine organism identification based solely on spectral signatures for small size fauna, without having a reliable visual verification (e.g., from RGB video), may raise concerns for organism abundance overestimation. [71] shows that the spectral signatures are sufficiently different to ensure the organism is present. Spectral signature information, however, did not improve estimations of macrofauna (4–15 cm) from those obtained from video data by human annotators. Yet, [71] explains that UHI datasets make benthic surveys more suitable for automation, especially for the fauna of highly variable appearance.

## 5. UHI Validation and Calibration: Pigment Extraction and Specimen Identification

The premise and effectiveness of UHI as a bio-optical tool able to identify organisms, monitor photosynthetic activity at small spatial scales ($mm^2$-$m^2$), and perform over other possible applications (as shown in Table 3), will rely heavily on the implementation of validation and calibration methods linking any seafloor biogeochemical features with their associated spectral signatures at adequate resolutions. With underwater RGB imagery, we are accustomed to most bio-optical applications to use visual annotations to either classify or predict features of interest (Figure 6). Instead, with UHI, we can develop more finely tuned relationships such as some of the reviewed applications targeting primary producers that have baselined proximal sensing estimates with extraction-based methods to quantify photosynthetic pigment content (e.g., Chl *a*). Few bio-optical studies have extracted pigments to explain the spectral signatures of benthic organisms for other classification studies (Table 2). This is understandable as, in ROV operations in the deep sea, for example, the surveys are often time-constrained, and sampling is logistically challenging.

**Table 3.** Current application capabilities of proximal hyperspectral imaging for marine ecosystems and organisms (adapted from [11]).

| Application | Microphytobenthos | Warm-Water Corals | Cold-Water Corals | Coralline Algae | Sponges | Oyster Reefs | Sympagic Environments | Mineral Resource Areas |
|---|---|---|---|---|---|---|---|---|
| Photosynthetic pigment content | Demonstrated | Lacking validation | N/A | Demonstrated | N/A | N/A | Demonstrated | Lacking validation |
| Species identification | Unproven | Demonstrated | Demonstrated | Unproven | Demonstrated | Unproven | Unproven | Demonstrated |
| Physiological assessments | N/A | Demonstrated | Demonstrated | Unproven | Unproven | N/A | Unproven | Unproven |
| *In situ* abundance | Demonstrated | Demonstrated | Demonstrated | Demonstrated | Demonstrated | Unproven | Demonstrated | Demonstrated |
| Epiphyte composition | N/A | Unproven | N/A | Demonstrated | Unproven | Demonstrated | N/A | N/A |
| Nutrient cycling | Demonstrated | Unproven | Unproven | Unproven | Unproven | Unproven | Unproven | Unproven |

"Demonstrated" are applications that have been validated and can become routinely applied in seafloor ecosystem research. "Lacking validation" are studies that have showcased a possible application but still require quantification and validation of the feature of interest. "Unproven" signals studies that have not been developed nor tested. "N/A" refers to that the feature of interest does not apply to the environment or organism.

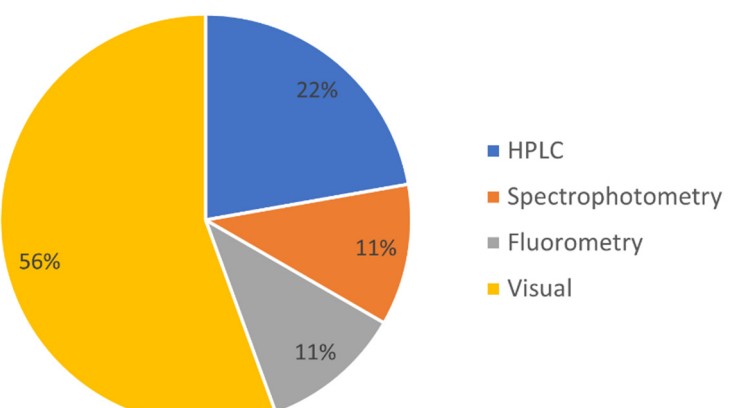

**Figure 6.** Summary statistics of validation or regression methods employed in UHI studies. Pigment extraction has been performed using high-performance liquid chromatography (HPLC), spectrophotometry, and fluorometry to derive pigment content such as Chl *a*. Visual refers to the manual annotation of seafloor objects or features of interest.

Validation protocols need to be carefully designed as extraction-based methods differ considerably between pigment types (e.g., chlorophylls, anthocyanins, phycobilins) and marine organisms, often requiring different chemical procedures or laboratory equipment [64,72,108,113]. High-performance liquid chromatography (HPLC) is considered the "gold standard" for estimating pigment concentration in marine algal organisms and is used for satellite hyperspectral observations of algal diversity or phytoplankton groups [120,121]. HPLC permits the separation of the constituents of a sample and resolves most of the different chlorophyll, carotenoids, and chlorophyllic degradation products that are unable to be identified via any other method (e.g., spectrophotometry) as they overlap in their absorption spectra [122]. This is important as the relative concentrations of photosynthetic and accessory pigments provide useful ecological, taxonomic and physiological information (Figure 5) [108]. However, when samples contain water-soluble (e.g., phycobilins) or unknown pigments, the costs of calibration alone can be a limitation. Thus, it is important to obtain *a priori* knowledge on the pigment composition in regards to the study objectives [64], then the research question, logistics, and funding limitations will determine the appropriate validation method. For example, [72] applied spectrophotometer analysis for water-soluble pigments and HPLC for non-polar pigments of benthic primary producers (see Section 4.3.1). On the other hand, [64] evaluated marker pigments that give different benthic fauna their characteristic colors via HPLC and liquid chromatography-mass spectrometry (see Section 4.4.).

HPLC and spectrophotometry methods provide similar results, especially in samples without Chl degradation products [122,123]. Therefore, HPLC validation is required, for example, for phytodetritus studies where the spectral signatures of Chl degradation products are part of the research question. For instance, in their deep-seafloor UHI research, [71] detected Chl *a* spectral absorbance. While no photosynthetic active radiation can reach the deep sea, [71] explain that accumulations of Chl *a* and corresponding degradation products (pheophytin *a* and pheophorbide *a*) have been detected at abyssal depths associated with phytodetritus deposition after seasonal phytoplankton blooms. These claims are supported by their UHI results where spectral signatures with minimum reflectance intensity around 668–680 nm suggest Chl *a* deposition, which would have been overlooked by RGB imagery. Pairing UHI with pigment-extraction methods capable of resolving Chl degradation products (e.g., HPLC) will increase our understanding of the ecological role of phytodetritus in the deep-sea and polar environments in a non-invasive manner. Further, spectral indices able to account for Chl *a* in degraded form (e.g., pheophytin), or other photosynthetic pigments present in seafloor sediments should be the focus of further research [124].

Regarding sediment or other natural substrates (e.g., such as sea-ice) an important feature to be considered during UHI data acquisition, processing, and validation is the

spatial and vertical variability of photosynthetic pigments present [27,76]. If the vertical spatial variability of photosynthetic microorganisms in their medium is not accounted for, hyperspectral data analysis with ancillary pigment extraction methods may differ by several tens of percent [27] or might yield an incorrect interpretation of the results [76]. An exemplar study case has been addressed ex-situ by [76] regarding microalgae vertical variability within the ice matrix. This behavior can be turned into an advantage coupled with UHI methods to monitor vertical migration changes/characteristics [27]. However, validation studies with different types of substrates (e.g., soft clays, sand), depth, and lightning-setups remain to be addressed [27].

Finally, we note that extraction-based methods such as HPLC may overlook certain pigments as they can occur below minimum detection levels [108,113]. Interestingly, [108] reports a higher sensitivity of hyperspectral imaging to detect certain photosynthetic pigments of algal groups than HPLC under certain circumstances. In addition, currently, all studies have focused on the visible spectrum, and [64] explains that the capability to detect pigments that absorb in the ultra-violet range and above 690 nm (Infra-red) wavelengths would require the development of more powerful light sources and/or sensitive UHI systems, to increase the opportunity to find species-specific pigment markers.

## 6. Discussion of Technical Challenges for UHI Systems for Seafloor Observations

There are key elements requiring attention before the widespread adoption of UHI as a valuable tool for seafloor ecosystem studies. Here, we identify some of the technical challenges for UHI systems that restrict its application and standardization as a method for benthic observations. These include sensor deployment and image acquisition aspects, such as (1) the effects of differences in survey altitude or illumination conditions in the accuracy of benthic classifications or regression models [67,73,74,125], (2) georeferencing issues related to positioning and orientation of the systems underwater [26,71,73], and (3) data analysis techniques capable of reducing the complexity of hyperspectral data cubes into useful information for seafloor ecosystem studies (Table 2).

### 6.1. Variable Survey Altitude and Uneven Illumination Effects

Compared to terrestrial application, the wavelength-dependent attenuation of light in underwater environments hampers UHI data quality and interpretation to a greater extent. The spectral data will inevitably be influenced by the inherent optical properties of the surrounding seawater, the illumination conditions on the target by natural or artificial light sources (e.g., sunlight or active lamps), the sensor distance to the seafloor, and the angularity of the field of view [26]. For example, [67] explain how their results were restricted by the lack of altitude corrections and attenuation of light in water, as spectral signatures of benthic organisms in deeper UHI transects were different due to more blue wavelengths reaching the sensor compared to the signal from red bands. In addition, although [67] provided a fixed depth value for spectral correction, they explain that unevenness in illumination increased the number of false negatives when classification algorithms were applied. Similarly, [73] found that illumination inconsistencies caused by lamps, ROV orientation, and canyon slope caused a high percentage of "dark" or shadow pixels. Therefore, careful consideration of light source and illumination effects, and maintaining a constant distance between the sensor to the area of interest, is critical to ensure high classification accuracies, particularly in heterogeneous habitats (e.g., coral reefs).

To preserve the accuracy of the spectral relationship between the area of interest and the variation in depth, [67] claimed that future studies should incorporate two additional measurements: (1) real-time altitude to the seafloor in the data acquisition log and (2) the water's in situ spectral attenuation coefficient. However, [74] claimed that by combining annotations from different altitudes, classification accuracy can be improved. Recent efforts by [125] sharing annotated UHI datasets are fostering the development of training algorithms and are developing an understanding of the effect of survey altitude on data quality. UHI studies require further research about the influence of minor altitude differences

caused by seafloor heterogeneity and complex benthic morphologies on classification accuracy [126]. Further, the coupling of digital bathymetric models obtained through imaging and acoustic devices with UHI data could be explored as a source to provide additional variables with which to correct or standardize the data [68,76].

*6.2. Navigation, Georeferencing, and Survey Procedures*

Since all current developed UHI systems employ a push-broom sensor architecture, which acquires pixel lines at high frequency to compose a transect image, they rely on accurate navigational data such as sensor attitude and positioning [68,78]. Some platforms such as the underwater motorized rails, or the under-ice sled, have avoided such requirements due to their stable and highly controlled movement. However, equipping UUVs (see Section 3.1.2) with navigation and dynamic positioning instruments is a necessary step forward to allow to increase the spatial extent of the surveys in a more efficient manner. The positioning also grants re-visiting survey locations, and thus techniques for geopositioning and image georectification need to be developed and applied as suggested in [68].

Overall, the appropriate navigation equipment and survey protocols will depend on the research question, the spatial scale of the process or feature to be analyzed, and the accuracy required. The dominant method for ROV positioning is acoustic baseline positioning, which presents in two types: ultra-short baseline (USBL) and long-baseline (LBL), which have their trade-offs but both increase the complexity of the operations and can be difficult to establish under certain circumstances [127,128]. The positioning accuracies achieved by USBL range from 1.5 to 10 m [129,130]. On the other hand, for shallow areas [67], there are advantages to equipping USV with its own real-time kinematic global positioning system (GPS) that can stay on the surface (without the need for acoustic underwater positioning) and inertial measurement unit for obtaining high-quality navigational data that retain the geometric accuracy obtaining georeferenced transects. The integration of multiple platforms, such as an AUV with a USV providing positioning and navigation corrections could enable unmanned geo-referenced hyperspectral mapping useful for long-term monitoring of the deep seafloor [131]. The possibility of more precisely georeferencing UHI data would increase the usefulness to larger spatial scale remote-sensing studies (e.g., satellite observations) and their temporal monitoring as a field validation tool [67]. Georeferencing is important as [74] and [67] explain that shallow UHI systems can be used as a field validation tool for space or aerial platforms.

We further noticed time-consuming survey procedures in UHI diver surveys, such as gray reference boards that need to be positioned underwater before acquiring data along transects. However, this differs slightly from standard diver-based RGB imaging benthic surveys, as these are already integrating seafloor control points (i.e., geodesic networks) where local coordinates (e.g., latitude, longitude, depth) are annotated to geo-reference benthic topographic models [132]. Similarly, [67] placed and photographed (as an RGB image) four wooden frames (40 × 40 cm) before the UHI survey being completed to validate the hyperspectral images, as well as three white metal sheets, which were fixed to the seafloor to be used as a reference spectra and to delimit the study area. However, [67] explain that the spectral reference board can be excluded from survey procedures as they applied two different types of spectral corrections, one based on the reference spectra (i.e., metal-sheet) and the other based on the average values from the UHI data, finding no differences between classification results.

An interesting avenue of research would be to use other types of hyperspectral image acquisition, like "snap-shot" sensors that capture an image similar to a normal camera with a fixed number of pixels in width and height. These have proven useful for studying camouflage in marine organisms [115,116]. As such, an underwater snap-shot design would be useful in fixed benthic installations for time-lapse analysis of seafloor communities, phytodetritus biogeochemistry, and degradation [51], as observations would not be affected by a changing survey altitude; however, they would require more powerful light sources.

*6.3. UHI Data Processing*

UHI data cubes are generally more demanding to analyze than RGB images due to the considerable amount of additional information in the spectral dimension ($\lambda$) (Figure 2) [26]. Although storage capacities are being overcome through technology advancement and miniaturization, hyperspectral image analysis will typically require a dimensionality reduction as most of the data will be correlated or can be redundant [14]. This is a sensitive process requiring different stages and assessment of optimal analysis techniques to retrieve biogeophysical features of the seafloor that will need to be explored in the underwater environment compared to terrestrial applications.

In general, the processing of UHI data requires a series of steps to be followed, including data pre-processing, georeferencing, segmentation, feature extraction, and data analysis. First, in the pre-processing step, UHI data has to be converted to radiance values (W m$^{-2}$ sr$^{-1}$ nm$^{-1}$) per pixel (i.e., radiometric correction). However, [27] used raw digital counts and reference panels to convert to reflectance values directly, bypassing the need to convert to spectral radiance. Nonetheless, radiometric correction can be beneficial as it allows the documentation of actual light levels, sensor intercomparison, and conversion to reflectance using additional tertiary sensors (thus avoiding panel allocation in some circumstances). The second step involves georeferencing, which places each pixel in a spatial arrangement and coordinate system through either the data from the integrated positioning and altitude sensors or spatial co-registration (see Section 6.2.). Finally, segmentation is employed to delimit areas within the image where the spectral analysis will be performed. This is an important step that excludes objects from the analysis that could affect the classification result.

Segmentation involves the selection of "regions of interest" (ROI) within the UHI transect (areas or targets from which spectral signatures will be retrieved). The selection of ROI is an important step as it aims to exclude irrelevant features within the image from the analysis. Currently, for UHI, ROI selection is still subjective, as most studies have selected ROIs from "manual" or RGB visual detection, which may carry bias in natural environments as it assumes that visual inspection will provide all that there is to classify. Spectral signatures from ROI will suffice as an input for further analysis; however, it is often required for spectral signatures to be interpreted as a vector for extracting information. In general, spectral indexes, principal component analysis, or double derivatives have been used for reducing the dimensionality of hyperspectral data for further analysis. The lack of ROI selection standard is a significant gap that should be addressed in future studies using UHI. Spectral homogeneity analysis [119] and deep learning methods [133] could aid in reducing ROI selection bias.

Although the study objectives will determine the type of analysis required, future studies could start exploring combinations of different spectral techniques synthesized in Table 2. In regards to data analysis, both for regression and classification, there is a plethora of algorithms and techniques that could be derived from proximal remote-sensing studies in terrestrial ecosystems to be adapted to the underwater environment, such as functional traits and diversity [56,134,135].

## 7. Conclusions

Our review of proximal UHI applications has showcased its potential for providing ecosystem attributes over a wide range of environments, allowing us to fill a niche gap in the spatial scales relevant for improved monitoring of impacts to marine ecosystems. Proximal hyperspectral imaging of marine habitats has provided information at spatial resolutions ranging from the sub-mm to the cm-scale of the seafloor composition and theoretically permits observations at different temporal resolution (minutes–hours–days) of benthic communities non-invasively, allowing the mapping of their immediate response to environmental cycles and impacts. Compared to RGB images, UHI holds significant advantages for seafloor habitat surveys as (1) it can provide more detailed information about seafloor surface biogeochemical properties and processes, (2) it fosters the automation

of benthic organism identification through their pigment's specific absorption, which shapes their unique spectral signature, and (3) it increases the detection of small fauna and flora not visible to human annotators. Challenges for future researchers will be to establish and validate different methods of acquiring and then translating UHI data into ecological and physiological information relevant for multi-disciplinary marine ecosystem monitoring and management.

**Author Contributions:** Conceptualization, J.C.M.-H., E.C. and V.L., investigation, J.C.M.-H., E.C.; writing—original draft preparation J.C.M.-H.; writing—review and editing, all authors.; visualization, J.C.M.-H.; supervision, V.C., A.L., N.H. and V.L.; project administration, V.L.; funding acquisition, V.L. All authors have read and agreed to the published version of the manuscript.

**Funding:** This research and J.C.M-H.'s Ph.D. scholarship were supported by the Australian Research Council's Special Research Initiative for the Antarctic Gateway Partnership (Project ID SR140300001).

**Institutional Review Board Statement:** Not applicable.

**Informed Consent Statement:** Not applicable.

**Data Availability Statement:** Not applicable.

**Conflicts of Interest:** The authors declare no conflict of interest. The funders had no role in the design of the study; in the collection, analyses, or interpretation of data; in the writing of the manuscript; or in the decision to publish the results.

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
