# Peer review of "Underwater Hyperspectral Imaging (UHI): A Review of Systems and Applications for Proximal Seafloor Ecosystem Studies"

_remotesensing, doi:10.3390/rs13173451_

Round 1

Reviewer 1 Report

In this paper, the authors reviewed the proximal UHI applications. In general, the authors just presented the work in various papers. That is to say, this paper lacks of authors’ own ideas. At this point, the novelty of this paper is plain. The authors should highlight their own work. The authors should discuss and analyze various methods from different papers. The advantages and disadvantages must be focused in detail. With this operation, the work in this paper would significantly help readers in the same fields.

  1. Many tables such as Table 1 and Table 2 listed in this paper. However, these parameters are just summarized and rewritten from various papers. The authors should discuss and analyze the advantages and disadvantages, which are the authors’ work. These work would guide the readers’ studies.

Author Response

Thank you for your constructive comments, we believe this has taken our manuscript in a better direction. This manuscript intends to synthesize disparate ideas, the relationships between each study, as well as chart a course of how the field can develop. We kept a review in mind staying to the current progress in the field, and partially including our own perspectives. Yet we acknowledge the reviewer's comment for the potential of this review to guide future studies. We have added further discussion points in regards to Table I and II, tradeoffs between systems and applications. Finally, we have made major changes to the manuscript, made it more polished, succinct, and comprehensive for the reader.

Reviewer 2 Report

The paper represents a comprehensive description of the current developments of Underwater Hyperspectral Imaging and its application for benthic habitat mapping. I would like to see this paper published after considering some minor comments listed below.

line 177: you may consider adding the one reference regarding underwater photogrammetry for archaeology:

Pydyn, A.; Popek, M.; Kubacka, M.; Janowski, Ł. Exploration and reconstruction of a medieval harbour using hydroacoustics, 3-D shallow seismic and underwater photogrammetry: A case study from Puck, southern Baltic Sea. Archaeological Prospection 2021, 10.1002/arp.1823, doi:10.1002/arp.1823.

lines 225-228: Can you provide the real positioning accuracies of USBL devices in the field? Is it possible to improve USBL measurement accuracy by increasing the number of USBLs? One or two sentences in this regard would improve this part of the manuscript.

Useful references:

Savini, A.; Vertino, A.; Marchese, F.; Beuck, L.; Freiwald, A. Mapping cold-water coral habitats at different scales within the Northern Ionian Sea (Central Mediterranean): an assessment of coral coverage and associated vulnerability. PLoS One 2014, 9, e87108, doi:10.1371/journal.pone.0087108.

lines 314-317 (Figure 4): annotations on Figure 4b are too small and difficult to read, the size of Figure 4f is too small.

lines 730-735: Consider supplementing the discussion of the possible application of UHI on an integrated UAV-USV system. See the reference: 

Zwolak, K.; Wigley, R.; Bohan, A.; Zarayskaya, Y.; Bazhenova, E.; Dorshow, W.; Sumiyoshi, M.; Sattiabaruth, S.; Roperez, J.; Proctor, A.; Wallace, C.; Sade, H.; Ketter, T.; Simpson, B.; Tinmouth, N.; Falconer, R.; Ryzhov, I.; Elsaied Abou-Mahmoud, M. The Autonomous Underwater Vehicle Integrated with the Unmanned Surface Vessel Mapping the Southern Ionian Sea. The Winning Technology Solution of the Shell Ocean Discovery XPRIZE. Remote Sensing 2020, 12, 1344, doi:10.3390/rs12081344.

Author Response

We appreciate your constructive comments, we believe this has taken our review to a better position for the underwater remote sensing community. Here are our responses to each comment.

  • Line 177: We don't think this reference is relevant to this section as we are excluding archeological studies using hyperspectral imaging. Yet the data fusion of Multi-Beam and Photogrammetry is of big interest to the development of this field and this reference will be taken into consideration.

- Line 225-228: USBL comment. 
Couldn't find the accuracies as only Johnsen et al. 2013 describes the possibility of the application but has not been used yet. However, we found USBL accuracies in the article provided by the reviewer and in Price et al. (2019). These accuracies have been added to the text line 822 - 824.

"Mean georeference error was 1.53 m (± 0.07 m SE), which is within error estimates of USBL navigation (1% of depth), and is relative to geographical location rather than within-model error (range 0.7–3 mm)".

- Price, D. M., Robert, K., Callaway, A., Lo lacono, C., Hall, R. A., & Huvenne, V. A. I. (2019). Using 3D photogrammetry from ROV video to quantify cold-water coral reef structural complexity and investigate its influence on biodiversity and community assemblage. Coral Reefs, 38(5), 1007–1021. https://doi.org/10.1007/s00338-019-01827-3

In the paper the reviewer provides, they say this "Absolute GPS-based positioning was performed using the shipboard IXSEA POSIDONIA USBL positioning system, reaching absolute accuracy in the range of 5–10 m."

- Figure 4 has been adjusted.

Lines 730 - 735: UAV-USV system possible integration.
This is an interesting application that could facilitate deep UHI surveys. Discussed in line 828.

Reviewer 3 Report

The manuscript presents a review of different applications and systems of hyperspectral imaging for marine environments. It presents an excellent literature review regarding the scope of the manuscript. I found the paper to be well written and described. There is no doubt that this review paper will add value to the scientific  community and is on a topic of relevance and general interest to readers of the journal. Therefore, I would recommend the manuscript for publication.

Author Response

Thanks for your comments, we appreciate the point of view provided and motivate us to continue our research.

Round 2

Reviewer 1 Report

The paper has been improved after revision. The reviewer suggests that this paper is accepted.

Author Response

We appreciate your comments and revision.